# TENSOR METHODS TO LEARN THE GREEN'S FUNCTION TO SOLVE HIGH-DIMENSIONAL PDE

## ABSTRACT

The method of Green's function plays an important role in solving PDEs. Recently deep learning models have been used to explicitly learn the Green's function to parameterize solutions of PDEs. DecGreenNet uses low-rank decomposition of the Green's function to obtain computational efficiency by separated learning from training data and Monte-Carlo samples. However, learning from a large number of Monte-Carlo samples for a high-dimensional PDE can lead to slow training and large memory requirements. As a solution we investigate on learning the Green's function by using tensor product grids generated by random partitions of dimensions. We propose DecGreenNet-TT by applying tensor-train structured low-rank decomposition to the Green's function and replace its components with neural networks that learn from partitions of each dimensions instead of all grid elements. We further propose DecGreenNet-TT-C to learn with a reduced number of neural networks by combining dimensions to generate combined tensor product grids. We further show that for the special case of separable source functions the Green's function can be constructed without multiplication of all tensor-train component neural networks leading to memory and computational efficiency. Using several Poisson equations we show that the proposed methods can learn with a collection of smaller neural networks compared to DecGreenNet to efficiently parameterize solutions with faster training times and low errors.

## 1 INTRODUCTION

In recent years, research on deep learning models for solving scientific problems are gaining rapid popularity. In particular, learning to parameterize partial-differential equations (PDE) using deep learning models such as Physics-Inspired-Neural-Networks (PINNs)(Dissanayake & Phan-Thien, 1994; Raissi et al., 2017) have gained a lot of attention due to their applications in many scientific domains such as fluid dynamics (Raissi et al., 2017) and molecular dynamics (Razakh et al., 2021). Compared to solving PDE problems with traditional numerical methods, use of deep learning models have open many possibilities such as parameterization of classes of PDEs (Anandkumar et al., 2019) and domain shift for PDE problems (Goswami et al., 2022). Despite many deep learning methods showing successful parameterization of solutions of two-dimensional PDE problems, effective parameterization for high-dimensional PDE problems remains to be explored further.

The method of Green's function is a well-known approach to solve and analyze PDEs (Evans, 2010; Courant & Hilbert, 1989). The use of deep learning models to learn Green's function is gaining popularity due its ability as a operator networks to parameterize both a specific instance of a PDE and a class of PDE (Anandkumar et al., 2019; Luo et al., 2022). DeepGreen (Gin et al., 2021) uses an auto-encoder model to learn Green's functions associated with boundary value problems. Boullé & Townsend (2022) and Boullé & Townsend (2023) used low-rank representations and sampling methods of learn Green's functions for elliptic and parabolic PDEs. These methods tends to solve a specific instances of a PDE. A more versatile approach to parameterize solution of a PDE is by using neural operators (Anandkumar et al., 2019), which parameterize Green's functions in various multi-layers architectures. Neural operators does not learn the underlying Green's function of a PDE and their scalability for high-dimensional PDEs are not well-known. Simultaneous learning of the Green's function and the parameterization solution of a PDE have been explored in recently proposed MOD-net (Luo et al., 2022) and GF-net (Teng et al., 2022). Both MOD-net (Luo et al., 2022) and GF-net (Teng et al., 2022) parameterize the Green's function with a neural network and minimize

residual and boundary losses similar to a PINNs. Hence, these models are more easier to interpret as well as to extend to parameterize different types of PDE problems.

Both MOD-net (Luo et al., 2022) and GF-net (Teng et al., 2022) apply Monte-Carlo integration to the parameterize the Green's function. A limitation with the Monte-Carlo integration is that it may require a large number of Monte-Carlo samples to solve high-dimensional PDEs. This could lead to computationally expensive training due to repetitive Green's function approximations and large memory requirements to store Monte-Carlo samples. Wimalawarne et al. (2023) proposed DecGreenNet to apply low-rank decomposition (Bebendorf & Hackbusch, 2003) to the Green's function by using two neural networks, where one network learns on the training samples and the other learns on the Monte-Carlo samples. DecGreenNet learns more efficiently compared to MOD-net since its neural networks can learn batch-wise from training samples and Monte-Carlo samples prior to feature multiplication to construct the Green's function. In spite of the success with two-dimensional problems, DecGreenNet may still face limitations with high-dimensional PDEs since learning from large number of Monte-Carlo samples would require large neural networks and large memory requirements.

In this paper, we explore computational and memory efficient methods to parameterize the Green's function for high-dimensional PDEs. We propose to learn the Green's function from a grid generated by tensor products of partitions from each dimensions. By applying tensor-train decomposition to the Green's function we construct models that learn from partitions of dimensions instead of learning from the whole collection of grid elements. We demonstrate that the resulting models learn from a collection of small neural networks to simulate the a grid leading to memory efficiency and faster training times. Through evaluations with several Poisson problems we show that our methods achieve high-accuracy, faster learning, and scalability for high-dimension PDEs.

**Notation** Tensors are multi-dimensional arrays and we represent a $K$-mode tensor as $A \in \mathbb{R}^{n_1 \times \cdots \times n_K}$. By notation $A[i_1, \ldots, i_K]$ we present the element of $A$ at indexes $i_1 \in [n_1], \ldots, i_K \in [n_K]$. The notation : on a dimension indicates all elements of that dimension. The reshape function reshapes a tensor $A \in \mathbb{R}^{n_1 \times \cdots \times n_K}$ to a tensor with dimensions $B \in \mathbb{R}^{m_1 \times \cdots \times m_{K'}}$ as $B = \text{reshape}(A, (m_1, \ldots, m_{K'}))$. We also use a roll function to rearrange tensors by shifting their dimensions, e.g. given $A \in \mathbb{R}^{n_1 \times n_2 \times n_3}$ a single shift in dimensions is $\text{roll}(A, 1) \in \mathbb{R}^{n_3 \times n_1 \times n_2}$. For a collection of vectors $\{y_1, y_2, \ldots, y_L\} \in \mathbb{R}^m$, the stack function sequentially stack each vector to generate a matrix as $\text{stack}(y_1, y_2, \ldots, y_L) \in \mathbb{R}^{L \times m}$.

## 2 LEARNING WITH THE GREEN'S FUNCTION

In this section, we briefly review the Green's function and neural network models that learn the Green's function to parameterize PDE problems. Let us consider a domain $\Omega \subset \mathbb{R}^N$, a differential operator $\mathcal{L}$, a source function $g(\cdot)$ and a boundary condition $\phi(\cdot)$, and present a general PDE problem as

$$\mathcal{L}[u](x) = g(x), \qquad x \in \Omega \tag{1}$$
$$u(x) = \phi(x), \qquad x \in \partial\Omega.$$

For simplicity, let us restrict to a linear PDE with Dirichlet boundary condition $\phi(\cdot) = 0$. Then, there is a Green's function $G : \mathbb{R}^N \times \mathbb{R}^N \to \mathbb{R}$ for a fixed $x' \in \Omega$ as follows:

$$\mathcal{L}[G](x) = \delta(x - x'), \qquad x \in \Omega$$
$$G(x, x') = 0, \qquad x \in \partial\Omega,$$

which leads to a solution function $u(\cdot)$ as

$$u(x; g) = \int_\Omega G(x, x') g(x') dx'. \tag{2}$$

Given a set $S_\Omega$ consisting of random samples from $\Omega$, the Monte-Carlo approximation of the Green's function integral 2 is

$$u(x; g) \approx \frac{|\Omega|}{|S_\Omega|} \sum_{x' \in S_\Omega} G(x, x') g(x'). \tag{3}$$

## 2.1 MOD-NET

Luo et al. (2022) proposed MOD-Net to parameterize the Green's function $G(x, x')$ in 3 using a neural network $G_{\theta_1}(\cdot, \cdot)$ with parameters represented by $\theta_1$ to formulate the following solution

$$u_{\theta_1}(x; g) = \frac{|\Omega|}{|S_\Omega|} \sum_{x' \in S_\Omega} G_{\theta_1}(x, x')g(x'). \tag{4}$$

Luo et al. (2022) demonstrated that MOD-Net can be parameterize solutions of a class of PDE for varying $g(\cdot)$, hence, behaves similar to a neural operator. They considered $K$ different $g^{(k)}(\cdot)$, $k = 1, \cdots, K$ to represent a class of PDE. Let $S^{\Omega,k}$ and $S^{\partial\Omega,k}$ are domain elements from the interior and boundary, respectively, for $k = 1, \ldots, K$. Then, the objective function of MOD-Net for the parameterized solution in 4 is given as

$$R_S = \frac{1}{K} \sum_{k \in [K]} \left( \lambda_1 \frac{1}{|S^{\Omega,k}|} \sum_{x \in S^{\Omega,k}} \|\mathcal{L}[u_{\theta_1}(x; g^{(k)})](x) - g^{(k)}(x)\|_2^2 \right.$$
$$\left. + \lambda_2 \frac{1}{|S^{\partial\Omega,k}|} \sum_{x \in S^{\partial\Omega,k}} \|u_{\theta_1}(x; g^{(k)})\|_2^2 \right), \tag{5}$$

where $\lambda_1$ and $\lambda_2$ are regularization parameters.

A limitation of learning with the Monte-Carlo approximation 4 is that the number of Monte-Carlo samples needed to approximate the Green's function would increase as the dimensions of the PDE increases. This will lead to computationally expensive large number of repetitive summations in 4 for each input $x$.

## 2.2 DECGREENNET

It has been well-established that a low-rank decomposition of a Green's function for elliptic (Bebendorf & Hackbusch, 2003) and parabolic PDEs (Boullé et al., 2022) can be established with two functions $F : \mathbb{R}^N \to \mathbb{R}^r$ and $G : \mathbb{R}^N \to \mathbb{R}^r$ such that $G(x, y) \approx F(x)^\top G(y)$ for some rank $R$. Wimalawarne et al. (2023) proposed DecGreenNet by using two neural networks $F_{\gamma_1} : \mathbb{R}^N \to \mathbb{R}^r$ and $G_{\gamma_2} : \mathbb{R}^N \to \mathbb{R}^r$ to learn the low-rank decomposed Green's function in 4 as

$$u_{\gamma_1, \gamma_2}(x; g) = F_{\gamma_1}(x)^\top \sum_{x' \in S_\Omega} H_{\gamma_2}(x')g(x'). \tag{6}$$

A nonlinear extension DecGreenNet-NL was also proposed with an additional neural network $O_{\gamma_3} : \mathbb{R}^P \to \mathbb{R}$ applied to the concatenation of the summed values in 6 as

$$u_{\gamma_1, \gamma_2, \gamma_3}(x; g) = O_{\gamma_3}\left( F_{\gamma_1}(x)^\top \mathrm{concat}\big[H_{\gamma_2}(y_1)g(y_1), \ldots, H_{\gamma_2}(y_P)g(y_P)\big] \right), \tag{7}$$

where $y_i \in S_\Omega$ and $P = |S_\Omega|$.

The main advantage of 6 and 7 is the separated batch-wise learning from training input elements and Monte-Carlo samples. Considerable gains in training times with DecGreenNet compared to MOD-Net have been reported for two-dimensions PDE problems(Wimalawarne et al., 2023). However, the scalability of DecGreenNet for high-dimension PDEs is not known.

## 3 PROPOSED METHODS

We propose to extend DecGreenNet for high-dimension PDE problems using tensor methods. Our proposal consists of two components (a) tensor product grid construction (b) tensor-train structured parameterization of the Green's function by neural networks. Lastly, for source functions that can be separated with respect to each dimension, we demonstrate a memory and computationally efficient restructuring of summations of tensor-train multiplications.

### 3.1 Tensor Product Grids

A common approach to create a grid is to use tensor products over segmentation of each dimension of the domain as used with sparse grids (Bungartz & Griebel, 2004) and integration methods (Vysotsky et al., 2021). We provide a general definition of the tensor product grid generation below.

**Definition 3.1** (Tensor Product Grid). *Let $x^{(a)} = \left\{ x_1^{(a)}, \ldots, x_p^{(a)} \right\}$ be $p$ (random) partitions of the dimension $a$. Then we define the tensor product grid by*

$$x^{(1)} \times x^{(2)} \times \cdots \times x^{(N)} = \left\{ \left( x_{j_1}^{(1)}, x_{j_2}^{(2)}, \ldots x_{j_N}^{(N)} \right) \ \middle| \ x_{j_1}^{(1)} \in x^{(1)}, \ \ldots, \ x_{j_N}^{(N)} \in x^{(N)} \right\}.$$

The above tensor product method may not scale well for high-dimensional PDEs since we need tensor products with respect to all dimensions. As an alternative, we propose to combine dimensions and apply tensor product on each combination, which we refer to as the *Combined Tensor Product Grid* and it's definition is given below.

**Definition 3.2** (Combined Tensor Product Grid). *Let us consider $N'$ disjoint sets from the $N$-dimensions $x^{(a)}$, $a \in [N]$ as $A_1, \ldots, A_{N'}$ with $A_i \cap A_j = \emptyset$. We consider $p_i$ (random) partitions of each set of dimensions by $X^{A_i} = \left\{ \left( x_j^{(a_1)}, \ldots x_j^{(a_{|A_i|})} \right) \middle| \forall a_k \in A_i, j = 1, \ldots, p_i \right\}$. Then, a grid construction can be constructed as*

$$X^{A_1} \times X^{A_2} \times \cdots \times X^{A_{N'}} = \left\{ \left( y_{j_1}^{(1)}, y_{j_2}^{(2)}, \ldots y_{j_{N'}}^{(N')} \right) \ \middle| \ y_{j_1}^{(1)} \in X^{A_1}, \ \ldots, \ y_{j_{N'}}^{(N')} \in X^{A_{N'}} \right\}.$$

To demonstrate the above construction, let $X = \{(0.2\,, 0.6), (0.5,\ 0.3)\}$ and $Y = \{0.4,\ 0.7\}$ then $X \times Y = \{(0.2\,, 0.6\,, 0.4), (0.2\,, 0.6\,, 0.7), (0.5\,, 0.3\,, 0.4), (0.5\,, 0.3\,, 0.7)\}$.

We want to emphasize that combining of dimensions and random partitioning for the combined tensor product grid need careful consideration since grid points may not be uniformly distributed. The benefit of this approach is that we can use fewer neural networks compared to the general tensor product grid for very high-dimensional PDEs. For now, we do not have any theoretically guarantees for combining dimensions and propose to consider the selection of the number of dimension combinations and their partitions by hyperparameter tuning for each PDE.

### 3.2 DecGreenNet-TT

Let us consider a domain $\Omega \subset \mathbb{R}^N$ where $N \geq 2$ with tensor product grids generated with each dimension $a$ having $p_a$ partitions of $x_{a_i}^{(a)} \in \mathbb{R}$, $a_i = 1, \ldots, p_a$. Let us consider a tensor product grid $x^{(1)} \times \cdots \times x^{(N)} := Q$ where $|Q| = p_1 p_2 \cdots p_N$. Now, all elements in $Q$ is needed to learn the DecGreenNet 6 as

$$u_{\gamma_1, \gamma_2}(x; g) \tag{8}$$

$$= F_{\gamma_1}(x)^\top \sum_{i_1=1}^{p_1} \cdots \sum_{i_N=1}^{p_N} H_{\gamma_2}\left( x_{i_1}^{(1)}, \ldots, x_{i_N}^{(N)} \right) g\left( x_{i_1}^{(1)}, \ldots, x_{i_N}^{(N)} \right)$$

$$= F_{\gamma_1}(x)^\top \left( \sum_{i_1=1}^{p_1} \cdots \sum_{i_N=1}^{p_N} \text{Roll}(\text{Reshape}\left( H_{\gamma_2}(\text{stack}(Q)), (p_1, \ldots, p_N, r) \right), 1)[:, i_1, \ldots, i_N] \right.$$

$$\left. g(x_{i_1}^{(1)}, \ldots, x_{i_N}^{(N)}) \right), \tag{9}$$

where the last line uses batch learning by $H_{\gamma_2}(\cdot)$ using all the tensor product grid elements with the input matrix $\text{stack}(Q) \in \mathbb{R}^{p_1 \cdots p_N \times N}$. The resulting $p_1 p_2 \cdots p_N \times r$ dimensional output of $H_{\gamma_2}(\text{stack}(Q)) \in \mathbb{R}^{p_1 \cdots p_N \times r}$ is reshaped and mode shifted to a tensor of dimensions $r \times p_1 \times p_2 \times \cdots \times p_N$. This naive extension of DecGreenNet has a memory requirement of $\mathcal{O}(Np^N)$ given that $p_1 = \ldots = p_N = p$ indicating that for high-dimensional domains learning from 9 is not computationally or memory efficient.

Now, we apply a decomposition method similar to tensor-train decomposition (Oseledets, 2009; Oseledets & Tyrtyshnikov, 2009) to $\text{Roll}(\text{Reshape}\,(H_{\gamma_2}(\text{stack}(Q)),(r,p_1,\ldots,p_N)),1)$, which gives the following decomposition

$$
\sum_{i_1=1}^{p_1}\cdots\sum_{i_N=1}^{p_N}\text{Roll}(\text{Reshape}\,(H_{\gamma_2}(\text{stack}(Q)),(r,p_1,\ldots,p_N)),1)\,[:,i_1,\ldots,i_N]g(x_{i_1}^{(1)},\ldots,x_{i_N}^{(N)})
$$
$$
\approx\sum_{i_1=1}^{p_1}\cdots\sum_{i_N=1}^{p_N}\left[\sum_{\alpha_1=1}^{r_1}\cdots\sum_{\alpha_{N-1}=1}^{r_{N-1}}G_1[:,i_1,\alpha_1]G_2[\alpha_1,i_2,\alpha_2]\cdots G_N[\alpha_{N-1},i_N,1]\right]g(x_{i_1}^{(1)},\ldots,x_{i_N}^{(N)})
$$
$$(10)$$

where $G_k\in\mathbb{R}^{r_k\times p_k\times r_{k+1}}$, $k=0,\ldots,N$ are component tensors with ranks $r_{k-1}$ and $r_k$, $i_0\in[r],i_1\in[p_1],\ldots i_N\in[p_N]$, $r_0=r$ and $r_N=1$.

The construction in 10 indicates that we can use separate neural networks to learn each $G_i$ $i=1,\ldots,N$ by using partitions $x^{(i)}$, $i=1,\ldots,N$ from each dimension. Hence, we propose to use $N$ neural networks

$$
T_{\theta_i}^{(r_{i-1},r_i)}:\mathbb{R}^{p_i\times 1}\to\mathbb{R}^{r_{i-1}\times p_i\times r_i},\ i=1,\ldots,N, \tag{11}
$$

which learn with respect to the partitions of size $p_i$ of the $i$-th dimensions. Using the 10 and 11, we propose the *DecGreenNet-TT* to replace 9 as

$$
u_{\gamma_1,\theta_0,\ldots,\theta_N}(x;g)=
$$
$$
F_{\gamma_1}(x)^\top\sum_{i_1=1}^{p_1}\cdots\sum_{i_N=1}^{p_N}\left[\sum_{\alpha_1=1}^{r_1}\cdots\sum_{\alpha_{N-1}=1}^{r_{N-1}}T_{\theta_1}^{(r_0,r_1)}(x^{(1)})[:,i_1,\alpha_1]T_{\theta_2}^{(r_1,r_2)}(x^{(2)})[\alpha_1,i_2,\alpha_2]\cdots\right.
$$
$$
\left.T_{\theta_N}^{(r_{N-1},1)}(x^{(N)})[\alpha_{N-1},i_N,1]\right]g\left(x_{i_1}^{(1)},\ldots,x_{i_N}^{(N)}\right). \tag{12}
$$

The above model still has a memory requirement of $\mathcal{O}(Np^N)$ given that $p_1=\ldots=p_N=p$ since it computes all summations on the left side similar to 12. However, DecGreenNet-TT could learn with a collection small neural networks since it learn only partitions of each dimension. On the other hand, the $H_{\gamma_2}(\cdot)$ network of DecGreenNet may require a large network to learn from all grid elements.

### 3.3 LEARNING FROM COMBINED TENSOR PRODUCT GRID

Learning $N$ neural networks to learn from partitions of each dimension in the previous formulation 12 can be redundant and computationally inefficient. Alternatively, we can combine dimensions to construct $N'<N$ and use Definition 3.2 to construct disjoint sets $A_1,\ldots,A_{N'}$ and generate $X^{A_1},\ldots,X^{A_{N'}}$ with partitions $X^{A_i}=\left\{\left(x_j^{(a_1)},\ldots x_j^{(a_{|A_i|})}\right)|\forall a_k\in A_i,j=1,\ldots p_i\right\}$.

Now, we can construct $N'$ neural networks as

$$
T_{\theta_i}^{(r_{i-1},r_i)}:\mathbb{R}^{p_i\times|A^i|}\to\mathbb{R}^{r_{i-1}\times p_i\times r_i},\ i=1,\ldots,N', \tag{13}
$$

where $r_0=r$, $r_N=1$, and $\theta_i$ indicates parameterized neural network. Combining 13 with the 12 lead to propose the *DecGreenNet-TT-C* as

$$
u_{\gamma_1,\theta_0,\ldots,\theta_{N'}}(x;g)=
$$
$$
F_{\gamma_1}(x)^\top\sum_{i_1=1}^{p_1}\cdots\sum_{i_{N'}=1}^{p_{N'}}\left[\sum_{\alpha_1=1}^{r_1}\cdots\sum_{\alpha_{N'-1}=1}^{r_{N'-1}}T_{\theta_1}^{(r_0,r_1)}(X^{(A_1)})[:,i_1,\alpha_1]T_{\theta_2}^{(r_1,r_2)}(X^{(A_2)})[\alpha_1,i_2,\alpha_2]\cdots\right.
$$
$$
\left.T_{\theta_{N'}}^{(r_{N'-1},1)}(X^{(X_{N'})})[\alpha_{N'-1},i_{N'},1]\right]g\left(x_{i_1}^{(1)},\ldots,x_{i_1}^{(|A_1|)},\ldots,x_{i_{N'}}^{(1)},\ldots,x_{i_{N'}}^{(|A_{N'}|)}\right). \tag{14}
$$

The above model still has a memory requirement of $\mathcal{O}(N'p^{N'})$ given that $p_1=\ldots=p_{N'}=p$ which can be smaller comapred to DecGreenNet-TT.

### 3.4 Memory Efficient Computation

In spite of the advantages of using multiple small networks and learning only by partitions of dimensions, 12 or 14 still have limitations of the storing a $r \times p_1 \cdots p_N$ or $r \times p_1 \cdots p_{N'}$ in memory and require a large number summation operations. We demonstrate that we can further improve memory efficiency for specific functions as we describe next.

**Definition 3.3** (Polynomial separable source functions). *A source function $g(x_1, \ldots, x_N)$ is called a polynomial separable source functions if it can be expressed as an element-wise multiplication of functions as $g(x_1, \ldots, x_N) = \sum_{k=1}^{K} g_1^{(k)}(x_1) g_2^{(k)}(x_2) \cdots g_N^{(k)}(x_N)$, where some $g_i^{(k)}(\cdot) = 1$.*

**Definition 3.4** (Fully separable source functions). *A source function $g(x_1, \ldots, x_N)$ is called a fully separable source functions if it can be expressed as an element-wise multiplication of functions as $g(x_1, \ldots, x_N) = g_1(x_1) g_2(x_2) \cdots g_N(x_N)$.*

An example of a polynomial separable source function is the source function used in the Poisson problem in (Luo et al., 2022) with $g(x_1, x_2) = x_1(x_1 - 1) + x_2(x_2 - 1)$. The Poisson problems used in (Teng et al., 2022) with solutions $u(x_1, x_2) = \sin(2\pi x_1)\sin(2\pi x_2)$ and $u(x_1, x_2) = \cos(2\pi x_1)\cos(2\pi x_2)$ lead to fully separable functions. We want to mention that not all functions can be constructed as separable function, i.e. $f(x_1, x_2) = \log(x_1 + x_2)$, $f(x_1, x_2) = \sqrt{(x_1^2 + x_2^2)}$, in such instance only 12 can be used.

It can be shown that separable source functions can improve memory and computational efficiency of DecGreenNet-TT and DecGreenNet-TT-C. We provide the following theorem to obtain a rearrangement of 10 for polynomial separable source function.

**Theorem 1.** *Let us consider a polynomial separable source function $g(x_{i_1}^{(1)}, \ldots, x_{i_N}^{(N)}) = \sum_{k=1}^{K} g_1^{(k)}(x_{i_1}^{(1)}) g_2^{(k)}(x_{i_2}^{(2)}) \cdots g_N^{(k)}(x_{i_N}^{(N)})$. Then, the 10 can be simplified as*

$$
\sum_{i_1=1}^{p_1} \cdots \sum_{i_N=1}^{p_N} \left[ \sum_{\alpha_1=1}^{r_1} \cdots \sum_{\alpha_{N-1}=1}^{r_{N-1}} G_1[:, i_1, \alpha_1] G_2[\alpha_1, i_2, \alpha_2] \cdots G_N[\alpha_{N-1}, i_N, 1] \right] g(x_{i_1}^{(1)}, \ldots, x_{i_N}^{(N)})
$$

$$
= \sum_{k=1}^{K} \sum_{i_1=1}^{p_1} \sum_{\alpha_1=1}^{r_1} g_1^{(k)}(x_{i_1}^{(1)}) G_1(:, i_1, \alpha_1) \sum_{i_2=1}^{p_2} \sum_{\alpha_2=1}^{r_2} g_2^{(k)}(x_{i_2}^{(2)}) G_2(\alpha_1, i_2, \alpha_2) \cdots
$$

$$
\sum_{i_N=1}^{p_N} g_N^{(k)}(x_{i_N}^{(N)}) G_N(\alpha_{N-1}, i_N, 1). \quad (15)
$$

Theorem 1 indicates sequential summations from the last tensor-train components towards the first component without computing with all tensor-train components. This will completely remove the need to construct the full $H_{\gamma_2}(\text{stack}(Q))$ and store in memory prior to multiplication with the source function $g(\cdot)$. Furthermore, each $\sum_{i_i=1}^{p_i} \sum_{\alpha_i=1}^{r_i} g_i^{(k)}(x_{i_i}^{(i)}) G_i(\alpha_{i-1}, i_i, \alpha_i)$ reduces to a vector of dimension $r_{i-1}$. Hence, memory requirement with is $\mathcal{O}(pr^2)$ where $p = \max\{p_1, \ldots, p_N\}$ and $r = \max\{r_0, \ldots, r_{N-1}\}$, which is considerable small compared to learning fully tensor structured models 10 or 14. By replacing $G_i(\alpha_{i-1}, i_i, \alpha_i)$ with $T_{\theta_i}^{(r_{i-1}, r_i)}(x^{(i)})[\alpha_{i-1}, i_1, \alpha_i]$, we can obtain the memory efficient DecGreenNet-TT 12 and also easily extended to DecGreenNet-TT-C 14.

## 4 Experiments

In this section, we parameterize standard PDE problems to evaluate accuracy and memory efficient learning of our proposed methods.

### 4.1 Experimental Setup

We experiments with DecGreenNet-TT and DecGreenNet-TT-C for both 2D problems and high-dimensional Poisson equations. We used PINNs (Raissi et al., 2017), DecGreenNet and DecGreenNet-NL (Wimalawarne et al., 2023) as a baseline method for all PDEs. All models use $\text{ReLU3}()$ as the activation function with last layer without any activation.

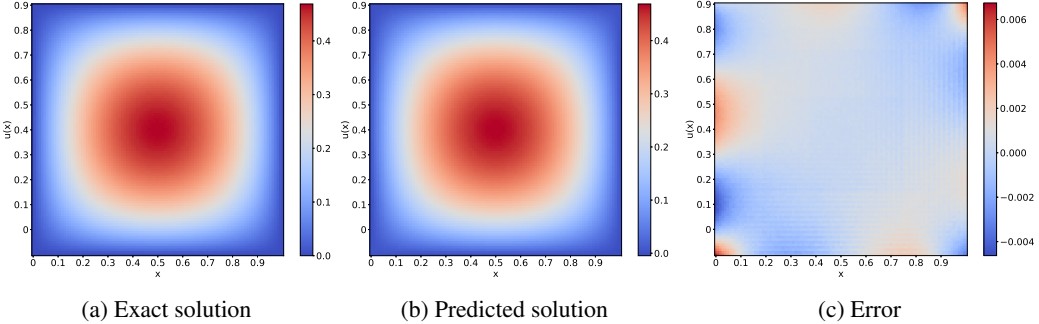

(a) Exact solution          (b) Predicted solution          (c) Error

Figure 1: Comparison between exact solution and DecGreenNet-TT for Poisson 2D equation for the interpolated solution at $a = 15$ with (a) exact solution, (b) solution by DecGreenNet-TT and (c) error.

All methods require several hyperparameters that need to be evaluated to select the optimal model. We represent network structures for all methods with notation $[in, h, \ldots, h, out]$, where $in$, $out$, and $h$ represent dimensions of input, output, and hidden layers, respectively. The hidden layers dimension $h$ is a hyperparameter selected from $2^h$ $h = 5, \ldots, 10$. All models require multi-layer networks where the layers are selected from $2, \ldots, 5$. For low-rank models, ranks of the Green's function is another hyperparameter selected, however, due to large number of possibilities we selected experimentally specified ranks depending on the problem. We selected the number of partitions in each dimension for DecGreenNet-TT and DecGreenNet-TT-C from the set $\{10, 100, 200\}$. The number of Monte-Carlo samples for DecGreenNet, and DecGreenNet-NL were selected to match number of grid samples used with the proposed methods or the maximum number of Monte-Carlo samples we can evaluate by experiments. Regularization parameters in 5 are $\lambda_1 = 1$ and $\lambda_2$ is selected from $1, 10, 100$. We used the Pytorch environment with Adam optimization method with learning rate of 0.001 and weight decay set to zero. We used a NVIDIA A100 GPUs with CUDA 11.6 on a Linux 5.4.0 environment to perform all our experiments. We provide code on experiments in this section at `https://github.com/tophatjap/DecGreenNet-TT`.

### 4.2 POISSON 2D PROBLEM

Our first experiment is Poisson 2D problem proposed in (Luo et al., 2022) given as

$$-\Delta u(x, y) = g^{(k)}(x, y), \quad (x, y) \in \Omega,$$
$$u(x, y) = 0, \qquad (x, y) \in \partial\Omega.$$

where $g^{(k)}(x, y) = -a_k(x^2 - x + y^2 - y)$ which has an analytical solution of $u(x, y) = \frac{a_k}{2} x(x - 1)y(y - 1)$. We experimented with both single PDE instance learning with $k = 1$ and operator learning to learn a class of PDEs a varying set of $k$ as in (Luo et al., 2022), (Wimalawarne et al., 2023).

We experimented with operator learning capability of the DecGreenNet-TT and DecGreenNet-TT-C by considering setting in (Wimalawarne et al., 2023) where multiple Poisson 2D equations with $a_k = 10k$ for $k = 1, \ldots, 10$ are solved simultaneously. Once a model learns the paramterization for the whole class of PDE, then the interpolated solution to unseen $a = 15$ is obtained. Figure 1 shows the interpolated solution and the error with respect to the analytical solution DecGreenNet-TT model with networks structures $F_{\gamma_1} = [2, 512, 512, 512, 512, 3]$, $T_1^{(r_0, r_1)} = [1, 64, 64, 64, 6]$, and and $T_2^{(r_1, 1)} = [1, 64, 64, 2]$ with $r_0 = 3$ and $r_1 = 2$. The low-error in the learned solution indicates strong learning capability of the proposed method to parameterize solution to multiple PDEs and its functionality of operator learning. Experiments to parameterize a single instance of a above Poisson equation with $a = 15$ were also conducted. As shown in Table 1 DecGreenNet-TT has obtained a competitive accuracy with respect DecGreenNet with a faster training time.

### 4.3 HOMOGENEOUS POISSON EQUATION

We constructed high-dimensional Poisson equations by extending the homogeneous Poisson equation studied in (Teng et al., 2022). The solution of the homogeneous 2D Poisson equation (Teng et al.,

| Method | Network structure | MC samples (Segments) | $\lambda_b$ | Error | Time (sec) |
|---|---|---|---|---|---|
| PINNs | [2, 64, 64, 1] | - | 10 | $1.07 \times 10^{-3}$ | 51.59 |
| DecGreenNet | $F_{\gamma_1}$ = [2,128, 128, 128, 5] $H_{\gamma_2}$ = [2, 16, 16, 16, 5] | 100 | 10 | $7.79 \times 10^{-4}$ | 185.49 |
| DecGreenNet-NL | $F_{\gamma_1}$ = [2, 256, 256, 5] $H_{\gamma_2}$ = [2, 32, 32, 32,5] $O_{\gamma_3}$ = [100,8,1] | 100 | 1 | $8.13 \times 10^{-3}$ | 138.49 |
| DecGreenNet-TT | $F_{\gamma_1}$ = [2,512,512,6] $TT_{\theta_1}^{(6,3)}$ = [1, 64, 64, 64, 18] $TT_{\theta_2}^{(3,1)}$ = [1, 64, 64,3] | 100 ($10^2$) | 10 | $4.90 \times 10^{-4}$ | 55.05 |

Table 1: Results for learning Poisson 2D equation for $a = 15$.

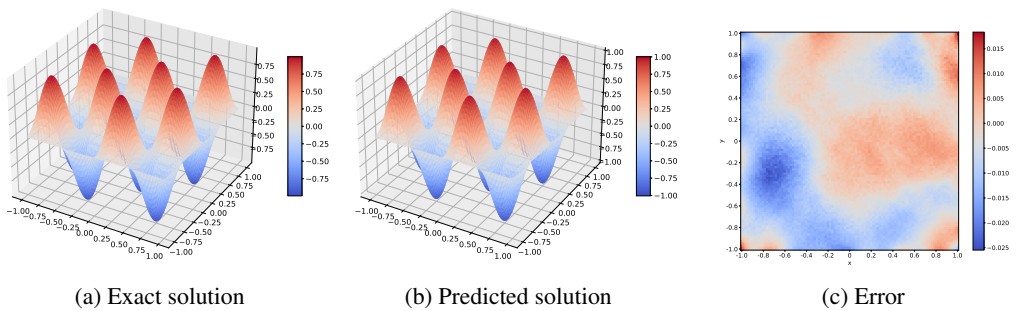

(a) Exact solution        (b) Predicted solution        (c) Error

Figure 2: Comparison between exact solution and DecGreenNet-TT for homogenous Poisson 2D equation with (a) exact solution, (b) solution solution and (c) error.

2022) in the domain of $\Omega = [-1, 1]^2$ is given by

$$u(x, y) = \sin(2\pi x)\sin(2\pi y), \qquad (16)$$

which we extend for $N$-dimensions by $u(x_1, \ldots, x_N) = \prod_{i=1}^{N} \sin(2\pi x_i)$. We also extended the above equation 16 to a general $N$-dimension PDE for a domain $\Omega \subset [-1, 1]^N$ by the following formulation

$$\sum_{i=1}^{N} \frac{\partial^2 u(x_1, \ldots, x_N)}{\partial x_i^2} = g(x_1, \ldots, x_N), \quad (x_1, \ldots, x_N) \in \Omega,$$

$$u(x_1, \ldots, x_N) = 0, \qquad (x_1, \ldots, x_N) \in \partial\Omega,$$

where $g(x_1, \ldots, x_N) = -4N\pi^2 \sin(2\pi x_1)\cdots\sin(2\pi x_N)$. Notice that $g(x_1, \ldots, x_N)$ is a fully separable function, which allows us to use memory efficient computations in section 3.4 in all our experiments. For dimension $N = 2, 3$ and $4$ we generated training sets of sizes 10000, 30000, and 80000, respectively from the interior of the domain. For the same dimensions, validation sets and test set sizes are 3000, 9000, and 24000. For each boundary, we generated 500 training samples, 300 validation samples, and 300 test samples.

Figure 2 shows comparison of the solution by DecGreenNet-TT to the exact solution of the homogeneous Poisson 2D equation. The network structure used consists of $F_{\gamma_1} = [2, 256, 256, 256, 256, 6]$, $T_1^{(r_0, r_1)} = [1, 64, 64, 18]$, and and $T_2^{(r_1, 1)} = [1, 64, 64, 3]$ with $r_0 = 6$ and $r_1 = 3$. The low error indicates that the proposed methods is capable of obtaining highly accurate solution.

We experimented with varying dimensions of $N = 2, 3, 4$ with DecGreenNet, DecGreenNet-TT, and DecGreenNet-TT-C and their results are given in Tables 3 and 2. For $N = 2, 3$, we used DecGreenNet-TT that learn from random partitions of 10 from each dimension to simulate 100 and 1000 grid elements, respectively. For $N = 4$, we used a DecGreenNet-TT-C by combining each two dimensions with random partitions of 200 from each combination to simulate a grid with 40000

| N | Network Structure | Segments (Samples) | $\lambda_b$ | Error | Time (sec) |
|---|---|---|---|---|---|
| 2 | $F_{\gamma_1} = [2, 256, 256, 256, 3]$ 
 $T_{\theta_1}^{(3,2)} = [1, 64, 64, 6]$ 
 $T_{\theta_2}^{(2,1)} = [1, 64, 64, 2]$ | 10 
 $(10^2 = 100)$ | 10 | 0.02259 | 58.92 |
| 3 | $F_{\gamma_1} = [3, 512, 512, 512, 512, 12]$ 
 $T_{\theta_1}^{(12,6)} = [1, 128, 128, 72]$ 
 $T_{\theta_1}^{(6,6)} = [1, 64, 64, 64, 36]$ 
 $T_{\theta_2}^{(6,1)} = [1, 64, 64, 6]$ | 10 
 $(10^3 = 1000)$ | 1 | 0.34718 | 195.76 |
| 4 | $F_{\gamma_1} = [4, 512, 512, 512, 32]$ 
 $T_{\theta_1}^{(32,6)} = [2, 256, 256, 132]$ 
 $T_{\theta_2}^{(6,1)} = [2, 128, 128, 128, 6]$ | 200 
 $(200^2 = 40000)$ | 1 | 0.2923 | 672.62 |

Table 2: Results on learning homogeneous Poisson equations varying dimensions (N) using DecGreenNet-TT and DecGreenNet-TT-C

| N | Network structure | MC samples | $\lambda_b$ | Error | Time (sec) |
|---|---|---|---|---|---|
| 2 | $F_{\gamma_1} = [2, 128, 128, 128, 128, 128, 5]$ 
 $H_{\gamma_2} = [2, 64, 64, 64, 64, 5]$ | 100 | 10 | 0.0147 | 90.25 |
| 3 | $F_{\gamma_1} = [3\ 512, 512, 512, 512, 512, 10]$ 
 $H_{\gamma_2} = [3, 256, 256, 256, 10]$ | 1000 | 1 | 0.7859 | 372.44 |
| 4 | $F_{\gamma_1} = [4, 512, 512, 512, 512, 5]$ 
 $H_{\gamma_2} = [4, 128,128,128,128, 5]$ | 2000 | 1 | 0.5682 | 1787.68 |

Table 3: Results on learning homogeneous Poisson equations varying dimensions (N) using Dec-GreenNet

elements. Table 2 shows that proposed models can learn with large grids compared to DecGreenNet as the dimensions increases. We found that finding neural networks for DecGreenNet for a larger grid beyond 2000 elements by hyperparameter tuning practically difficult. It is important to notice in Table 3 that large grids require large networks for $H_{\gamma_2}$ of DecGreenNet. However, since DecGreenNet-TT and DecGreenNet-TT-C learn only on partitions of dimensions to simulate grids, network structures of $T_*^{(r_{*-1}, r_*)}$ are smaller compared to $H_{\gamma_2}$. The ability to learn with a collection of small neural networks has allowed both memory efficiency as well as faster learning with proposed methods. Furthermore, our proposed methods has obtained a smaller error compared to DecGreenNet for high-dimensions.

We want to emphasize from Tables 2 and 1 that first network of tensor-train components $T_{\theta_1}^{(r_1, r_2)}$ is large compared to other tensor-train component neural networks. Further, we found that in general the shared rank $r_1$ between $F_{\gamma_1}$ and $T_{\theta_1}^{(r_1, r_2)}$ needs to be large compared to other ranks of the tensor-train structure. We believe that these observations are due to feature sharing among the $F_{\gamma_1}$ and the tensor train component networks $T_*^{(r_{*-1}, r_*)}$. Furthermore, we observed that the network of $F_{\gamma_1}$ becomes bigger as the dimensions of the PDE increases, perhaps, to learn from the increased number of input training samples.

## 5 CONCLUSIONS AND FUTURE WORKS

We present DecGreenNet-TT and DecGreenNet-TT-C by applying tensor train based decomposition to parameterize the Green's function to learn from partitions of dimensions to simulate tensor product grids. We further show that for separable source functions the proposed methods can obtain memory efficiency by rearrangement of summation of tensor train components and avoiding construction of full Green's function. Through experiments we verified our claims that the proposed methods obtain faster training times and accuracy compared to DecGreenNet for high-dimension PDEs. We believe that extension of existing theory (Bebendorf & Hackbusch, 2003) of low-rank decomposition of the Green's function to tensor-train decomposition is an important future research.

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
