# OpenReview forum: "Tensor methods to learn the Green's function to solve high-dimensional PDE"
_ICLR.cc/2024/Conference — Submitted to ICLR 2024_

### Official Review · Reviewer_Ympv · 2023-10-19

**Soundness:** 2 fair
**Presentation:** 1 poor
**Contribution:** 1 poor
**Rating:** 3
**Confidence:** 3

**Summary:**

This paper proposes the DecGreenNet-TT-C algorithm for learning the Green’s function to solve high-dimensional PDEs. Their method is motivated by the DecGreenNet [1], which is based on the low-rank approximation of the Green’s function for elliptic and parabolic PDEs. The authors further generalize this approach by introducing a tensor-train parametrization of the Green’s function. Each of the tensor in the tensor-train decomposition is again parametrized with a neural network. The notion of combined tensor product grid is proposed to improve efficiency. Experiments on the 2D Poisson and homogeneous Poisson are conducted to demonstrate the performance of the method.

[1] Wimalawarne, Kishan, Taiji Suzuki, and Sophie Langer. "Learning Green's Function Efficiently Using Low-Rank Approximations." arXiv preprint arXiv:2308.00350 (2023).

**Strengths:**

- The paper has a clear motivation. The introduction, related works, and the problem settings are well-written and easy to follow.
- The DecGreenNet-TT-C method is a natural extension to the previous work [1].
- The experiment part is detailed.

**Weaknesses:**

- The presentation needs improving. Some notations are confusing, e.g. $|\Omega|$ and $|\S_\Omega|$ in (3), using $N$ to denote the dimension rather than the sample count. I suggest the authors use \eqref instead of \ref. $S^{\Omega,k}$ uses the superscript while $S_\Omega$ uses the subscript. $G_{\gamma_2}$ is defined above (6) but $H_{\gamma_2}$ is used within. “it’s” on page 4.
- See Questions.

**Questions:**

1. According to Table 1, it seems there is no reduction in the number of samples required. With the same amount of samples and the same neural network architecture for the Green function, how does DecGreenNet-TT-C compare to the original algorithm DecGreenNet? Does the structure of the samples enforced by the TT framework affect the performance?
2. On Page 6, I don’t see why (12) can be used to express functions like $\log(x_1 + x_2)$ and $\sqrt{x_1^2 + x_2^2}$. Also, can you explain the intuition why the Green function has a low-rank structure so that the TT can be adopted? And if not, what is the benefit of this method compared to MOD-Net [2]?
3. Is the training stable? Do the gradient-based optimization method work for the landscape of this particular form involving the nonlinearity and ill-posedness of the TT?

[2] Zhang, Lulu, et al. "MOD-Net: A machine learning approach via model-operator-data network for solving PDEs." arXiv preprint arXiv:2107.03673 (2021).

---

> ### Author Response · Authors · 2023-11-22
>
> Thank you for the valuable comments.
>
> - For the first experiment, we used same number of Monte-Carlo sample for all models, hence, Table 1 shows same sample size for DecGreenNet and DecGreenNet-TT. If you look at the Table 2 and Table 3, you can see that DecGreenNet-TT can learn with a large number of Monte-Carlo samples with higher accuracy and lower training times for higher-order PDE compared to DecGreenNet. This is possible since DecGreenNet-TT learns from partitions of dimensions with tensor-products via tensor-train decomposition instead of storing a large number of Monte-Carlo samples in as input for DecGreenNet.
>
> - Yes, it is possible to approximate $\log(x_1 + x_2)$ and $\sqrt{x_1 + x_2}$ by a polynomial of $x_1$ and $x_2$. However, such expansions will require many neural networks to construct the tensor-train structured Green’s functions as in Theorem 1. Such approach can be computationally expensive and may lead to low accuracy. The directly separable functions as given in the paper can be easily integrated with the proposed model.
>
> - We do not have a theoretical analysis for the tensor-structured low-rankness for Green’s function. Establishing tensor low-rankness is a future work and in this paper we have made an assumption that it exists. We point out that robust learning of PDEs in experiments justifies our claim that low-rankness exists for the Green’s function.
>
> - MOD-net is computationally expensive to train for 2D PDE problems due to approximation of the Monte-Carlo integral approximation as demonstrated in [1]. The low-rank decomposition of the Green’s function has shown to improve the training times with DecGreenNet. In this paper we further improve on the DecGreenNet with tensor-structured low-rank decomposition and tensor-product grids (inspired by sparse grid and Smolyak grid [2])
>
> - Training is stable for the problems we have used in the paper.
>
> [1] Wimalawarne, Kishan, Taiji Suzuki, and Sophie Langer. "Learning Green's Function Efficiently Using Low-Rank Approximations." arXiv preprint arXiv:2308.00350 (2023)
>
> [2] Hans-Joachim Bungartz and Michael Griebel. Sparse grids. Acta Numerica, 13:147–269, 2004.

---

### Official Review · Reviewer_572E · 2023-10-30

**Soundness:** 2 fair
**Presentation:** 2 fair
**Contribution:** 2 fair
**Rating:** 3
**Confidence:** 4

**Summary:**

This paper proposes novel tensor-based methods to efficiently learn Green's functions for high-dimensional PDEs. The key innovation is applying tensor train decomposition to represent the Green's function using separate neural networks that learn from partitions of dimensions. This compositional approach allows learning with multiple small networks instead of a single large one. For separable sources, they further rearrange summations to avoid constructing the full tensor, providing memory savings. Experiments on Poisson equations demonstrate their methods achieve higher accuracy and faster training than baseline techniques.

**Strengths:**

The authors provide a reasonable motivation for developing more efficient methods to learn parametrized Green's functions for high-dimensional PDEs. The usage of the tensor-train and combined tensor product grid to reduce the computation complexity of the Green function is interesting and reasonable.

**Weaknesses:**

1. The presentations should be largely improved. There are many typos and vague things that make some parts of the paper hard to follow. For example:

- in eq6, should $H_{\gamma_2}$ be $G_{\gamma_2}$?

- for the paragraph under eq12 to explain, it says "since it computes all summations on the left side similar to 12" -- should it be "similar to equation 9"?

- for the Table 2 in the experiment part, should the "Segments" for "N=4" be $200^4$ instead of $200^2$?

- still, for Table 2 in the experiment part, the caption said "results ..using
 DecGreenNet-TT and DecGreenNet-TT-C", but only one column "error" there -is the result of the  DecGreenNet-TT or DecGreenNet-TT-C?


2. The experiment's setting and results are relatively weak

- As the motivation targets for high-dim PDE, setting the maximum dim =4 with only the Poisson equation family is not convincing enough. Explorations with higher-order settings on more kinds of PDEs are encouraged.

- For the Poission with N=3,4 the error at the level of e-1, which is far away from the acceptable solution precision.

- I understand the trick of the Combined-Tensor-Product-Grid lacks a theoretical guarantee, but more numerical results should be posted with different partition settings with this trick, which helps to show how exactly this trick helps to improve the efficiency and how it could leverage the efficiency and model complexity.

3. The idea of applying Tensor-Train format to approximate function[1][2] is not novel. More discussions on the prior work are encouraged.

[1]:Bigoni, Daniele, Allan P. Engsig-Karup, and Youssef M. Marzouk. "Spectral tensor-train decomposition." SIAM Journal on Scientific Computing 38.4 (2016): A2405-A2439.

[2]:Gorodetsky, Alex A., Sertac Karaman, and Youssef M. Marzouk. "Function-train: a continuous analogue of the tensor-train decomposition." arXiv preprint arXiv:1510.09088 (2015).

**Questions:**

see weakness part

---

> ### Author Response · Authors · 2023-11-22
>
> Thank you for the valuable comments.
>
> - We extended existing Poisson problems to higher-orders in our experiments since we could not find a suitable higher-order PDE problem in the existing machine learning literature. Does the reviewer has any suggestions?
>
> - For N=4, we have used DecGreenNet-TT-C with combinations of 2 dimensions and from each combination we made 200 random partitions, hence, the total number of simulated samples are 200^2. All the other orders we used DecGreenNet-TT.
>
> - In the Table 2, for N=2,3 we have used DecGreenNet-TT since we do not combine any dimensions. This can be identified by the number of $T_{*}$ networks. For N=4, we combined dimensions by pairs, so we only have 2 networks of  $T_{1}$ and $T_{2}$, hence, we used the DecGreenNet-TT-C.
>
> - We will include suggested references in the paper and update the relationships to the proposed method.
>
> - We conducted further experiments with fixed equal spaced grid partitions for homogeneous Poisson problems with N=2 and N=3 using DecGreenNet-TT with similar networks structures used in Table 2. It shows that our proposed method performs well with different grid constructions.
>
> N |  Error    | Time (sec)
>
> 2  |  0.006     |    51.97
>
> 3  |  0.205     | 146.21

---

> > ### Comment · Reviewer_572E · 2023-11-22
> >
> > Thanks for the authors' response. I believe in the potential of the proposed work but still decided to hold the current score at this time.

---

### Official Review · Reviewer_Mq8o · 2023-10-31

**Soundness:** 3 good
**Presentation:** 2 fair
**Contribution:** 2 fair
**Rating:** 3
**Confidence:** 4

**Summary:**

The paper proposes new neural-network-based methods for learning the Green's function of a partial differential equation (PDE). The main focus of the paper is a Green's function approximation based on a tensor train network, with each site described by a neural network. The paper considers tensor product grids with uniform and random point structure. The experimental evaluation compares to a baseline method without tensor train factorization for both grid types.

**Strengths:**

+ The combination of tensor train with neural networks used to model the Green's function is likely novel.
 + The use of tensor train factorization shows improvements in runtime over the baseline method.

**Weaknesses:**

- The paper seems to refer to [Bebendorf and Hackbusch, 2003] to justify the claim that Green's matrices are approximable with low-rank. However, that paper and much other literature on approximation of kernel/Green's matrices is using *block* low rank structure of the Green's matrices. The low-rank structure arises as a result of e.g., interaction of far away particles, which is described by a block given by a subset of rows and columns of the Green's matrix. The overall Green's matrix is invertible. This is why H-matrices, HSS-matrices, and the butterfly decomposition all try to apply low-rank approximation on blocks and not the overall the matrix. The use of tensor train decomposition here makes sense, it is to an extent capturing the block low-rank structure. But the paper does not present things in this way and I believe claims that Green's matrices generally admit good low rank approximations to be false/misleading.
 - The paper does compare its approach to relevant literature beyond very brief qualifications in the introduction, prior to definition of the new method.
 - The combined tensor product grid, as far as I see from the mathematical definition, is just a tensor product grid with random as opposed to uniform spacing in each dimension. This can be presented in simpler terms.
 - The experimental evaluation seems to consider only very simple PDE model problems, without noise. These types of methods would be of interest to apply to experimental data to infer complex dynamics.
 - The experimental results do not compare to any other existing methods. It would also make sense to compare to a tensor train with plain tensors, instead of neural nets in place of tensors.
 - The paper uses python notation instead of mathematical notion (eq 10), and contains frequent minor wording errors.

Overall, I do not recommend the paper for publication in its current form, as I think the above weaknesses are major deficiencies that are not easily fixed.

**Questions:**

- Please clarify whether or not I correctly understood the definition of the combined grid (see weaknesses).

---

> ### Author Response · Authors · 2023-11-22
>
> Thank you for the valuable comments.
>
> - Method such as H-matrix and butterfly decomposition  are indeed used in solving PDE problems with Green’s function or Kernels [1]. In our understanding, for Green’s function formulation as ours can be formulated with Galerkin methods or collocation methods  (Page 117 [1]) and solved as a linear system where matrix inverse approaches can be employed as suggested by the reviewer.  If we parameterize the Green’s function with a neural network and apply Galerkin methods or collocation methods, then such model will not be equivalent to  the PINNs framework. We are not certain, however, such models may be more computationally expensive to train neural networks using back-propagation due to recursive multiplications used in methods such as H-matrix.
> We agree that our proposed methods of tensor-structured decomposition of the Green’s function does not stem from existing theory. Our objective in this paper is to propose a model using tensor-train based decomposition with combination of sparse grid and Smolyak grid [3] to parameterize the Green’s function and support it with experiments. We think that analytical methods in [2] can be extended to tensor structured decomposition of the Green’s function, which we propose as a future research.
>
> - Regarding notations and writing, we will update the writing in the revised version.
>
> - Our referenced PDE models [4] and [5] have not considered the noise in PDE experiments. Hence, we also did not consider noise based PDE problems.
>
> - In this paper, we wanted to propose the tensor-train model to parameterize the Green’s function and demonstrate its scalability with to higher-order PDEs. Hence, we used simple PDE problems to make our case.
>
>
>
> [1] Mario Bebendorf, Hierarchical Matrices, 2008
>
> [2] Bebendorf, M., Hackbusch, W. Existence of ℋ-matrix approximants to the inverse FE-matrix of elliptic operators with L ∞-coefficients. Numer. Math. 95, 1–28 (2003
>
> [3] Hans-Joachim Bungartz and Michael Griebel. Sparse grids. Acta Numerica, 13:147–269, 2004.
>
> [4] Zhang Lulu Luo, Tao Zhang Yaoyu, E Weinan, John Xu, Zhi-Qin, and Ma Zheng. Mod-net: A machine learning approach via model-operator-data network for solving pdes. CiCP, 2022
>
> [5] Yuankai Teng, Xiaoping Zhang, Zhu Wang, and Lili Ju. Learning green’s functions of linear reaction-diffusion equations with application to fast numerical solver. In MSML, 2022

---

### Official Review · Reviewer_KPfc · 2023-11-01

**Soundness:** 3 good
**Presentation:** 3 good
**Contribution:** 3 good
**Rating:** 5
**Confidence:** 3

**Summary:**

The authors highlight the significance of Green's function in solving PDEs and the recent utilization of deep learning models for learning and parameterizing solutions. While DecGreenNet offers computational efficiency through low-rank decomposition, the reliance on numerous Monte-Carlo samples for high-dimensional PDEs can lead to slow training and high memory requirements. To address these concerns, the authors propose DecGreenNet-TT, which incorporates tensor-train structured low-rank decomposition along with neural networks trained on partitions of each dimension. Additionally, they introduce DecGreenNet-TT-C, which efficiently combines dimensions to generate combined tensor product grids, thereby reducing the number of required neural networks. The proposed methods demonstrate improved training efficiency and low errors in parameterizing solutions compared to DecGreenNet, particularly when dealing with separable source functions.

**Strengths:**

This study provides an effective tool for addressing high-dimensional PDE problems. Specifically, this paper proposes a new method based on the Tensor Train decomposition to parameterize the Green's function and apply it to high-dimensional PDE solving. Comparative experiments with the DecGreenNet method demonstrate that the proposed DecGreenNet-TT and DecGreenNet-TT-C methods offer faster training time and higher accuracy.

**Weaknesses:**

The experimental evaluation should be extended to include a broader range of complex examples to validate the universality and effectiveness of the proposed methods.

**Questions:**

In numerical experiments, the authors do not show the comparison with the traditional method in computational mathematics. What is the baseline?

There are some typos when citing the equations.

Given a $N$, how to choose $N^{'}$ ?

---

> ### Author Response · Authors · 2023-11-21
>
> Thank you for the valuable comments.
>
> - One of the baseline methods we considered was MOD-Net, however, it requires a considerable computation time for higher-order PDEs. Hence, we did not consider MOD-Net as a baseline methods. For reference for the 2D Poisson problem we refer to the [1].
> We also considered PINNs as a baseline methods for homogeneous Poisson problem, which we list below.
>
>
> N  |  Network structure               |   $\lambda_b$   |    Error     |   Time (sec)
>
> 2   |   [2, 512, 512, 512, 1]          |   10               |   0.3428   |    13.18
> 3   |   [3, 512, 512, 512, 1]          |   1                 |   3.6025   |    9.60
> 4   |   [4, 512, 512, 512, 512, 1]  |   1                 |   36.71     |    13.59
>
> the above results show that PINNs perform weaker compared to the proposed methods.
>
>
> - There is not specific way that we can use for combining dimensions to find N'. We believe that we have select the best combination of dimensions by experiments. Hence, the the combinations of the dimensions and N' can be considered as hyperparameters.
>
> [1] Wimalawarne, Kishan, Taiji Suzuki, and Sophie Langer. "Learning Green's Function Efficiently Using Low-Rank Approximations." arXiv preprint arXiv:2308.00350 (2023)

---

### Meta-Review · Area_Chair_ivJo · 2023-12-10

**Metareview:**

The paper learns a Green function for a PDE by parametrizing it in form of the tensor train  The method is compared to previous approaches.

Strengths:
a) The idea is a natural extension of low-rank decomposition to multidimensional case
b) Numerical experiments are reasonable

Weaknesses:
The decomposition is not well-motivated. Its description is vague even for specialists in tensor decompositions.
For example, the formula (12) seems to be highly non-standard, and it is not even clear that it corresponds to tensor-train matrices (or matrix product state operators), which could be the natural extensions.

I think, theoretical study of the justification of the format should be done at least for simple cases like Poisson equation.

**Justification For Why Not Higher Score:**

The paper is written clearly and lacks motivation for the usage of the format.

**Justification For Why Not Lower Score:**

N/A

---

### Decision · Program_Chairs · 2024-01-16

Reject